# In Situ Study of Graphene Oxide Quantum Dot-MoSₓ Nanohybrids as Hydrogen Evolution Catalysts

**Marco Favaro** [1,†] [iD]**, Mattia Cattelan** [1,‡] [iD]**, Stephen W. T. Price** [2] [iD]**, Andrea E. Russell** [3] [iD]**, Laura Calvillo** [1,*] [iD]**, Stefano Agnoli** [1] **and Gaetano Granozzi** [1] [iD]

1   Dipartimento di Scienze Chimiche, Università di Padova, Via Marzolo 1, 35131 Padova, Italy;
    marco.favaro@helmholtz-berlin.de (M.F.); mattia.cattelan@bristol.ac.uk (M.C.);
    stefano.agnoli@unipd.it (S.A.); gaetano.granozzi@unipd.it (G.G.)
2   Finden Ltd., 1.12 Building R71, Harwell Campus, Oxfordshire OX11 0QZ, UK; stephen@finden.co.uk
3   School of Chemistry, University of Southampton, Highfield, Southampton SO17 1BJ, UK;
    A.E.Russell@soton.ac.uk
*   Correspondence: laura.calvillolamana@unipd.it
†   Present address: Helmholtz-Zentrum Berlin für Materialien und Energie GmbH, Institute for Solar Fuels,
    Hahn-Meitner-Platz 1, D-14109 Berlin, Germany.
‡   Present address: School of Chemistry, University of Bristol, Cantocks Close, Bristol BS8 1TS, UK.

**Abstract:** Graphene quantum dots (GOQDs)-MoSₓ nanohybrids with different MoSₓ stoichiometries (x = 2 and 3) were prepared in order to investigate their chemical stability under hydrogen evolution reaction (HER) conditions. Combined photoemission/electrochemical (XPS/EC) measurements and *operando* X-ray absorption spectroscopy (XAS) were employed to determine the chemical changes induced on the MoSₓ-based materials as a function of the applied potential. This in situ characterization indicates that both MoS₂ and MoS₃ materials are stable under operating conditions, although sulfur terminal sites in the MoS₃ nanoparticles are converted from S-dimer ($S_2^{2-}$) to S-monomer ($S^{2-}$), which constitute the first sites where the hydrogen atoms are adsorbed for their subsequent evolution. In order to complete the characterization of the GOQDs-MoSₓ nanohybrids, the composition and particle size were determined by X-ray photoemission spectroscopy (XPS), X-ray diffraction (XRD) and Raman spectroscopy; whereas the HER activity was studied by conventional electrochemical techniques.

**Keywords:** in line XPS-electrochemistry; operando XAS; HER

## 1. Introduction

The use of hydrogen as an energy carrier in fuel cells (FCs) is one of the most promising energy policies for a rapid transition from fossil fuels to more sustainable energy sources. In this context, it is mandatory to find clean and easy methods for hydrogen production. Most of the hydrogen used today is obtained by steam reforming of methane; however, the H₂ produced through this route contains a small amount of carbon monoxide, a poison for Pt-based electrocatalysts, that makes it unsuitable for its direct use in FCs. The production of hydrogen by electrochemical water splitting, exploiting the hydrogen evolution reaction (HER), is therefore gaining importance, also because this alternative production method allows the storage of the electricity intermittently generated by renewable sources and further reduces the dependency on hydrocarbon fuels. Since this reaction is catalytically activated, there is now renewed interest in the development of efficient HER electrocatalysts. Platinum-group-metals (PGMs) are the best candidates as HER electrocatalysts, but they are unsuitable to empower such technology on a large scale because they are considered critical raw materials [1,2]. Thence nowadays,

intense research activity is underway to test earth-abundant elements as HER catalysts, alternative to PGMs based ones, which are capable of operating in water, providing both high current density at a low overpotential, and sufficient current durability [3,4]. In this context, transition metal chalcogenides (TMCs) (e.g., $MoS_2$ and other sulfides and selenides) derived nanomaterials represent promising HER catalysts [3,5]. Nevertheless, the mechanism of HER activity for $MoS_x$-based catalysts is still disputed [6]. Most works agree on the key role played by sulfur species in the HER, however, the large variety of different structural building blocks, which are the basis of the many possible crystalline, amorphous or polymeric molybdenum sulfide phases, comprising $\mu$-$S^{2-}$/$\mu$-$S_2^{2-}$, terminal $\eta^2$-$S_2^{2-}$, apical $S^{2-}$, and unsaturated $S^{2-}$, makes the establishment of direct structure-activity relationships very difficult. Moreover, complex physicochemical changes can be triggered by the reducing electrochemical potential, making the interpretation of the experimental data even more complex. For example, some authors suggested that $S^{2-}$ monomer, typical of the Mo edge in $MoS_2$, form the most active sites [7]; on the other hand, other works outline the key role of $\eta^2$-$S_2^{2-}$ species [8]. It has been also suggested that the removal of sulfur species driven by the reducing potential can produce unsaturated Mo(IV)/Mo(V) species that, via the formation of transient metal hydrides, are the active sites in the evolution of hydrogen [9,10]. Very recently, combined experimental and theoretical data suggested that bridging $\mu$-$S^{2-}$ units connecting Mo(V) and Mo(IV) cations exhibit extremely high activity and that the presence of oxygen in the anion lattice consequent to the etching of sulfur species at reducing potential, is also very beneficial for electroactivity [11–13].

Given the complexity outlined above, experiments that allow the investigation of materials using *operando* conditions are essential to study the cathodic-triggered formation of $MoS_x$ species. In addition, these measurements should provide a good representation of the ultimate stability of the electrocatalysts under operating conditions. For example, several groups have reported that amorphous $MoS_3$ is not stable under the operating cathodic conditions [7–9], and it is gradually reduced to a sulfur deficient structure very similar to $MoS_2$. In contrast, other authors have suggested the presence of Mo(V) [9] or alternatively Mo(III) [8], however, the presence of unsaturated cation species seems to be a common feature.

In the present work, we describe a new synthesis method to prepare graphene quantum dot (nanometric graphene sheets, GOQD)-$MoS_x$ nanohybrids with different $MoS_x$ stoichiometries (x = 2 and 3). Graphene-based materials have emerged as promising platforms for growing $MoS_2$ due to their high surface area and good stability [12–14]. We have extended this approach to GOQD since the use of very small graphene sheets (<5 nm) is optimal for the development of bottom-up synthesis protocols, resulting in an intimate interaction between the two phases. The result is an advanced nanocomposite made up of small $MoS_x$ nanoparticles (NPs) with abundant exposed edge sites, highly dispersed and stabilized by a graphene matrix. We have used combined electrochemical/photoemission measurements, as well as *operando* X-ray absorption spectroscopy (XAS), to track the changes experienced by the material under operating conditions. This complete methodological approach allowed us to determine the actual $MoS_x$ species involved in the HER.

## 2. Materials and Methods

### 2.1. Synthesis and Physicochemical Characterization of GOQD-$MoS_x$ Nanohybrids

$(NH_4)_2MoS_4$ (4.3 mg) was dissolved in 1.6 mL of GOQDs solution (1 mg mL$^{-1}$) and sonicated for 20 min. Subsequently, 10 $\mu$L of Nafion solution (5 wt.% solids in alcohol and water, Sigma-Aldrich, Milan, Italy) was added to the suspension and sonicated for another 10 min. The synthesis of the GOQDs is detailed in the Supplementary Materials (SM).

For the XAS measurements, the electrodes were manufactured by painting the GOQD-$(NH_4)_2MoS_4$ suspension on to carbon paper (TGP-H-60, Toray Industries Inc., Otsu-shi, Japan) -to obtain a final Mo loading of 1.25 mg cm$^{-2}$. Circular button electrodes of 1.25 cm$^2$ area were cut and annealed at 250 °C and 700 °C in UHV to obtain the final GOQD-$MoS_3$ and GOQD-$MoS_2$ electrodes, respectively.

For the XPS/EC measurements, 80 μL aliquots of GOQD-(NH$_4$)$_2$MoS$_4$ suspension were drop casted on to a glassy carbon electrode and dried in an N$_2$ atmosphere. The electrodes were introduced into the UHV system and annealed at 250 °C and 700 °C to obtain the final GOQD-MoS$_3$ and GOQD-MoS$_2$ electrodes, respectively.

Raman spectra were acquired using a ThermoFisher DXR Raman microscope (Thermo Fisher Scientific, Milan, Italy). The spectra were recorded using a laser with an excitation wavelength of 532 nm (1 mW), focused on the sample with a 50× objective (Olympus Italia s.r.l., Milan, Italy).

The X-ray diffraction (XRD) characterization was performed with a Philips PW 1729 (Koninklijke Philips N.V., Amsterdam, The Netherlands), configured with a glancing angle geometry, operating with Cu Kα radiation (λ = 0.15406 nm) generated at 30 kV and 40 mA. The mean crystallite size was calculated from the MoS$_2$(002) peak using the Scherrer equation: Lc = k·λ/ß·cosθ, where k is the shape factor (k = 0.9), λ is the X-ray wavelength, ß is the line broadening at half the maximum intensity of the peak, and θ is the Bragg angle.

## 2.2. Electrochemical Measurements

The electrochemical measurements were conducted in a standard three-electrode electrochemical cell. A glassy carbon rod was used as a counter electrode and a saturated calomel electrode (SCE) placed inside a Luggin capillary was used as a reference electrode. A 0.5 M H$_2$SO$_4$ solution, prepared from high purity reagents (Sigma-Aldrich, Milan, Italy) and purged with argon gas, was used as supporting electrolyte. All the electrochemical experiments were carried out at room temperature. Ten cyclic voltammograms between +0.3 V and −0.1 V vs. RHE at 0.1 V s$^{-1}$ were performed to get good contact between the electrode and electrolyte and, subsequently, three more scans were measured at 0.020 V s$^{-1}$. Polarization curves were recorded between +0.2 and −0.3 V vs. RHE using a scan rate of 0.005 Vs$^{-1}$. The third linear sweep voltammetry (LSV) was used to compare the performance of the different catalysts toward the HER.

## 2.3. Operando X-ray Absorption Spectroscopy (XAS) Measurements

An in situ electrochemical cell was used to collect XAS data of the MoS$_x$-based catalysts as a function of the applied potential [15] (Figure S1a). The working electrode was held in place by an Au wire contact, a Pt wire served as the counter electrode, and the reference electrode was a mercury mercurous sulfate (Hg/Hg$_2$SO$_4$) electrode (calibrated as +0.697 V vs. RHE) that was connected to the cell via a short length of tubing containing the electrolyte. The cell was controlled by an Autolab potentiostat running with NOVA 1.11 Software (Methohm Autolab, Amsterdam, The Netherlands). The electrolyte was purged with N$_2$ and then pumped through the cell using a peristaltic pump. The samples were prepared as button electrodes (fresh electrodes) and the measurements were carried out at four different fixed potentials. Subsequently, the electrodes were subjected to an accelerated aging treatment (AAT) (aged electrodes), which consisted of 500 cycles between +0.2 V and −0.25 V vs. RHE at 0.050 Vs$^{-1}$, in order to study the stability of the samples and, then, XAS measurements were performed at the same fixed potentials. The measurements were run in 0.5 M H$_2$SO$_4$ purged with N$_2$. Prior to the XAS measurements, three cyclic voltammograms were collected between 0.3 and −0.1 V vs. RHE at 50 mV s$^{-1}$ to clean the surface and ensure full contact between the electrode and electrolyte. Three further cycles were recorded at 10 mV s$^{-1}$ to ensure that the electrode was stable. Linear scan voltammetry was run to reach the desired potential for the XAS measurements. X-ray absorption measurements were recorded on beamline B18 at Diamond Light Source (UK) with ring energy of 3 GeV and a current of 300 mA. The monochromator used was Si(311) crystals operating in Quick EXAFS (QEXAFS) mode. The measurements were carried out in fluorescence mode at the Mo K (19999 eV) absorption edge at 298 K using a 9-element Ge detector. Calibration of the monochromator was carried out using a Mo foil previously to the measurements. The acquired data were processed and analyzed using the programs Athena and Artemis [16], respectively, which implement the FEFF6 and IFEFFIT codes [17]. Fits were carried out using a *k* range of 3–16 Å$^{-1}$ for the MoS$_2$$^-$ and 3–12 Å$^{-1}$

for the $MoS_3^-$ based materials and an *R* range of 1.4–3.2 Å with multiple *k* weightings of 1, 2 and 3. Different FEFF inputs were used in these fits depending on the materials. A $MoS_3$ model was created based on data from References [18,19].

*2.4. In Line Photoemission and Electrochemical Measurements*

Measurements were performed in an ultrahigh vacuum (UHV) system that consists of two independent UHV chambers: a preparation/analysis chamber and an electrochemical (EC) chamber. In the preparation/analysis chamber, the samples were prepared using the procedure described above and, subsequently, characterized by X-ray photoemission spectroscopy (XPS). The UHV-EC transfer system, which consisted of two manipulators (horizontal and vertical), was connected to the main preparation chamber through a gate valve. The horizontal manipulator was used to transfer the sample from the analysis chamber to the EC chamber, whereas the vertical one allowed the sample to be raised to couple it to the electrochemical cell, which was connected to the EC chamber from the top. A custom made PEEK (polyether ether ketone) cell was used for the electrochemical measurements (Figure S1b). A Pt wire was used as counter electrode and an Ag/AgCl/Cl$^-$ (3M KCl) electrode placed in a Luggin capillary was used as reference electrode. The cell was controlled by an Autolab potentiostat running with NOVA 1.8 Software. A 0.1 M $HClO_4$ solution, prepared from high purity reagents (Sigma-Aldrich) and purged with argon gas, was used as the electrolyte. The electrolyte was pumped into the EC cell through a tubing system using a syringe pump (N-1010, Pump Systems Inc.), which allows accurate control of the flow. The electrolyte inlet consisted of a capillary tube (diameter ~0.35 mm) placed in the center of the cell, whereas the outlet is constituted by eight holes (diameter 0.5 mm) placed around the central capillary. Prior to the EC measurements, the tubing system was purged with Ar to remove the oxygen and then, it was filled with the electrolyte. The samples were polarized at two different potentials: (a) before HER (+0.18 V vs. RHE); (b) at HER conditions (−0.4 V vs. RHE). In this case, in order to see more significant changes in the samples, a more negative potential was selected for the measurement at HER conditions than in the XAS measurements, since the XPS data was not acquired simultaneously to the EC measurement. All the electrochemical experiments were carried out at room temperature, in an Ar atmosphere in order to avoid contact of the sample with oxygen, and using a flow rate of 1 mL min$^{-1}$.

The chemical changes induced by the electrochemical work were analyzed by XPS after each electrochemical measurement. Photoemission data were obtained in a custom designed UHV system equipped with an EA 125 Omicron electron analyzer (Scienta Omicron GmbH, Taunusstein, Germany) with five channeltrons, working at a base pressure of $10^{-9}$ mbar. Core level photoemission spectra (S 2p, Mo 3d, C 1s and O 1s regions) were collected at room temperature with a non-monochromatized Al Kα X-ray source (1486.7 eV) and using 0.1 eV steps, 0.5 s collection time and 20 eV pass energy.

## 3. Results and Discussion

The $MoS_x$ phase was determined by Raman spectroscopy and the stoichiometry by XPS (Figure 1a,b). Raman spectra of both GOQD-$MoS_x$ samples show the characteristic D (1350 cm$^{-1}$), G (1590 cm$^{-1}$), 2D (2700 cm$^{-1}$) and D + D'(2900 cm$^{-1}$) bands of graphene, confirming the presence of GOQDs [20]. Both commercial $MoS_2$ (Aldrich) and GOQD-$MoS_2$ show two bands at 380 cm$^{-1}$ and 405 cm$^{-1}$ that correspond to the in-plane $E^1_{2g}$ and out-of-plane $A_{1g}$ vibrational modes (Figure S2), respectively, which are characteristic of 2H-$MoS_2$ [21]. In the case of GOQD-$MoS_2$, the broadening and the intensity decrease of the $A_{1g}$ band can be associated with a small number of stacked layers along the *c* axis [22], confirming the limited growth of the $MoS_2$ NPs in presence of GOQDs. The broadening of the $E^1_{2g}$ band is related to the presence of in-plane defects sites, which contributes to the increase of active sites [23,24]. The Raman spectrum of GOQD-$MoS_3$, however, shows broad and weak bands due to the amorphous nature of the $MoS_3$ nanoparticles [25,26]. The broad bands at 282–382 cm$^{-1}$ correspond to the *v*(Mo-S) stretching and the band at 450 cm$^{-1}$ to the *v*($Mo_3$-$\mu_3$S) vibration. The bands

at 525 cm$^{-1}$ and 555 cm$^{-1}$ are attributed to S-S stretching from the terminal and bridging disulfide bonds $(S_2)^{2-}$, respectively, supporting the chain-like structure proposed for amorphous $MoS_3$ [9,25,26].

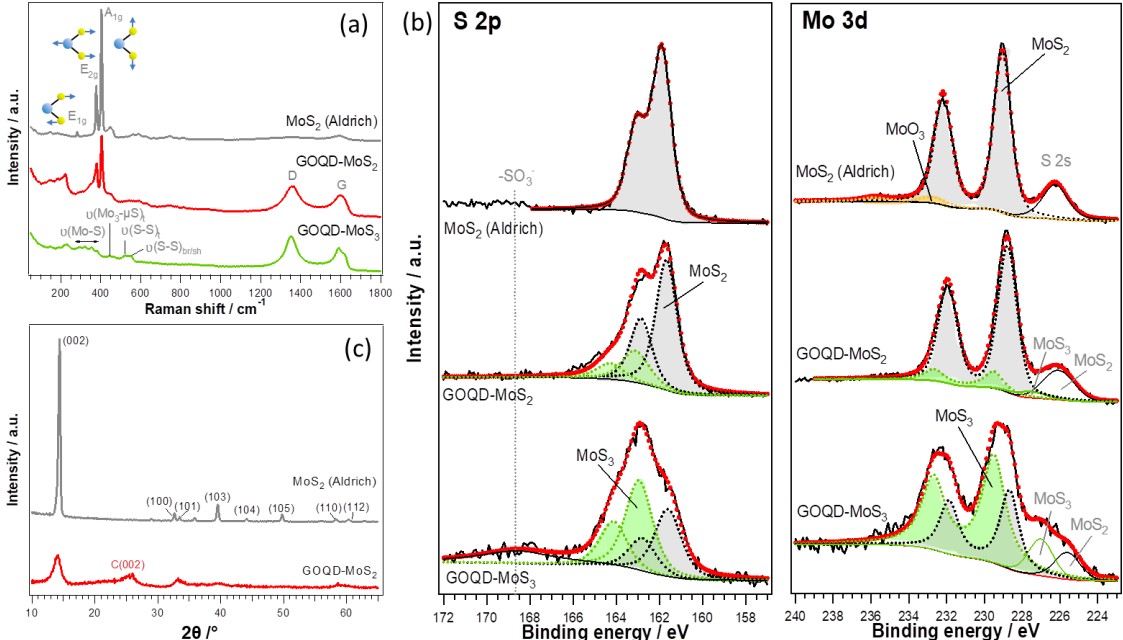

**Figure 1.** (**a**) Raman spectra; (**b**) S 2*p* and Mo 3*d*/S 2*s* photoemission spectra; and (**c**) XRD patterns for the GOQD-MoS$_x$ (x = 2 and 3) nanohybrids and the commercial MoS$_2$ (Aldrich). GOQD.

In the commercial MoS$_2$, the Mo 3$d_{5/2}$ peak is located at 229.0 eV and the corresponding S 2$p_{3/2}$ at 161.8 eV, which are characteristic of stoichiometric and crystalline MoS$_2$ [27,28]. The Mo 3$d_{5/2}$ photoemission (PE) line also shows a small component at a binding energy (BE) of 232.6 eV, attributed to the presence of MoO$_3$ on the edges due to oxidation by air exposure. GOQD-MoS$_2$ displays the characteristic features of MoS$_2$, confirming the success of the synthesis method; however, in this case, the peaks are slightly wider than in the case of the commercial MoS$_2$, suggesting the residual presence of a small amount of MoS$_3$ (components at 229.5 eV and 163.1 eV in the Mo 3$d$ and S 2$p$ PE, respectively). On the other hand, the Mo 3$d_{5/2}$ PE line of GOQD-MoS$_3$ presents two components at 228.8 and 229.5 eV related to Mo(IV) and Mo(V), respectively. The S 2$p_{3/2}$ PE line also shows two components at 161.8 eV, attributed to divalent sulfide ions (S$^{2-}$), and 163.0 eV, associated with μ-S$_2^{2-}$ and/or apical S$^{2-}$ [7,9]. The atomic Mo:S ratio calculated from the Mo 3$d$ and S 2$p$, taking into account the corresponding sensitivity factors, is 1:2.1 for GOQD-MoS$_2$ and 1:2.7 for GOQDs-MoS$_3$. It should be noticed that the BEs for the MoS$_2$ component in the commercial MoS$_2$, both in the Mo 3$d$ and S 2$p$ PE lines, are shifted 0.2 eV towards higher values respect to those for the GOQD-MoS$_x$ samples. This shift is attributed to the higher size of the commercial MoS$_2$ nanoparticles.

Figure 1c compares the X-ray diffraction (XRD) patterns of GOQD-MoS$_2$ and the commercial MoS$_2$. GOQD-MoS$_2$ shows six peaks at 2θ values of 14.1°, 32.9°, 39.5°, 49.7°, 58.4° and 60.3° which are attributed to the (002), (100), (103), (105), (110) and (112) reflections of the hexagonal structure of MoS$_2$ (seen in the XRD pattern of the commercial MoS$_2$) [29]. In addition, it shows a broad peak at 2θ = 25° associated with the presence of GOQDs [30,31]. The crystallite size of the MoS$_2$ NPs was determined from the XRD patterns by applying the Scherrer equation to the MoS$_2$(002) peak (Figure S2). The crystallite size calculated for MoS$_2$ and GOQD-MoS$_2$ was 26.5 nm (702 nm$^2$) and 7.4 nm (43 nm$^2$), respectively. This confirms once more that, in the presence of GOQDs, the growth of MoS$_2$ NPs is inhibited, leading to an abundance of exposed edge sites dispersed in the GOQDs matrix. GOQD-MoS$_3$ (not shown), however, did not show reflections, confirming its amorphous nature as seen by Raman spectroscopy. For this reason, it was not possible to determine the actual MoS$_3$ disordered NPs size;

however, we expect that it is similar to that of the MoS$_2$ NPs or even smaller (see EXAFS section) due to the lower temperature used in the synthesis.

The HER activity of the GOQD-MoS$_x$ nanohybrids was investigated in a standard three-electrode half-cell, using an Ar-saturated 0.5 M H$_2$SO$_4$ solution as electrolyte Figure 2 shows the polarization curves obtained for GOQD-MoS$_x$, as well as the corresponding Tafel plots. The results for the commercial MoS$_2$ and the carbon paper have also been included for comparison. Two different regions in the polarization curves are observed for all MoS$_x$-based materials. This effect has already been observed in the literature for MoS$_3$ [32]. In that case, the first region was attributed to the formation of reduced molybdenum sulfides, mainly MoS$_2$, whereas the second one was associated with hydrogen evolution. Considering the second process in order to compare the activity of the different materials toward HER, it can be observed that both GOQDs-MoS$_x$ exhibit a very similar activity, which is higher than that showed by the commercial MoS$_2$ in terms of overpotential. However, the three MoS$_x$-based materials show a very similar Tafel slope, indicating that the presence of GOQDs does not modify the kinetics of the process. The enhancement of the activity can be explained by the increase of exposed edge sites due to the coupling with GOQDs that limit the growth of the MoS$_x$ NPs, as confirmed by XRD.

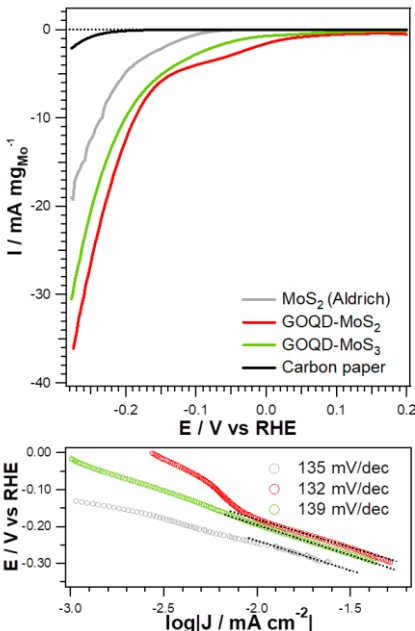

**Figure 2.** Polarization curves (not IR corrected, upper panel) and corresponding Tafel plots (bottom panel) in deaerated 0.5 M H$_2$SO$_4$ for the GOQD-MoS$_x$ (x = 2 and 3) and commercial MoS$_2$ modified electrodes acquired at room temperature and at a scan rate of 5 mV s$^{-1}$.

It is interesting to highlight the similar behavior of the GOQD-MoS$_x$ materials with different compositions. To further investigate the origin of the activity of these materials, *operando* XAS characterization was performed under HER conditions. The measurements were carried out at the Mo K edge to establish possible chemical changes as a function of the applied potential. For each material, four different potentials were studied: (i) before HER at +0.18 V$_{RHE}$; (ii) onset of the first slope in the polarization curve (+0.04 and −0.07 V$_{RHE}$ for GOQDs-MoS$_x$ and MoS$_2$, respectively); (iii) onset of the second slope in the polarization curve (−0.12 and −0.18 V$_{RHE}$ for GOQDs-MoS$_x$ and MoS$_2$, respectively); (iv) under HER conditions at −0.23 V$_{RHE}$. The measurements were performed on fresh and aged electrodes. Figure 3a–c shows the Fourier transformed extended X-ray absorption fine structure (FT EXAFS) spectra of GOQD-MoS$_x$ and commercial MoS$_2$, whilst the corresponding *k*-space spectra are reported in Figure S3. The MoS$_2$-based materials show two peaks at 2.0 and 2.8 Å (apparent distance), fitted to Mo-S and Mo-Mo interactions, respectively; whereas the MoS$_3$-based

sample exhibits the Mo-Mo interaction peak at 2.5 Å. No Mo-O contributions could be fitted for any of the samples, not even in the ex situ measured spectra (not shown). This does not exclude the presence of oxide on the edges; if present, as the fraction of Mo atoms that have O neighbors in the first coordination shell is too small to be reliably fitted.

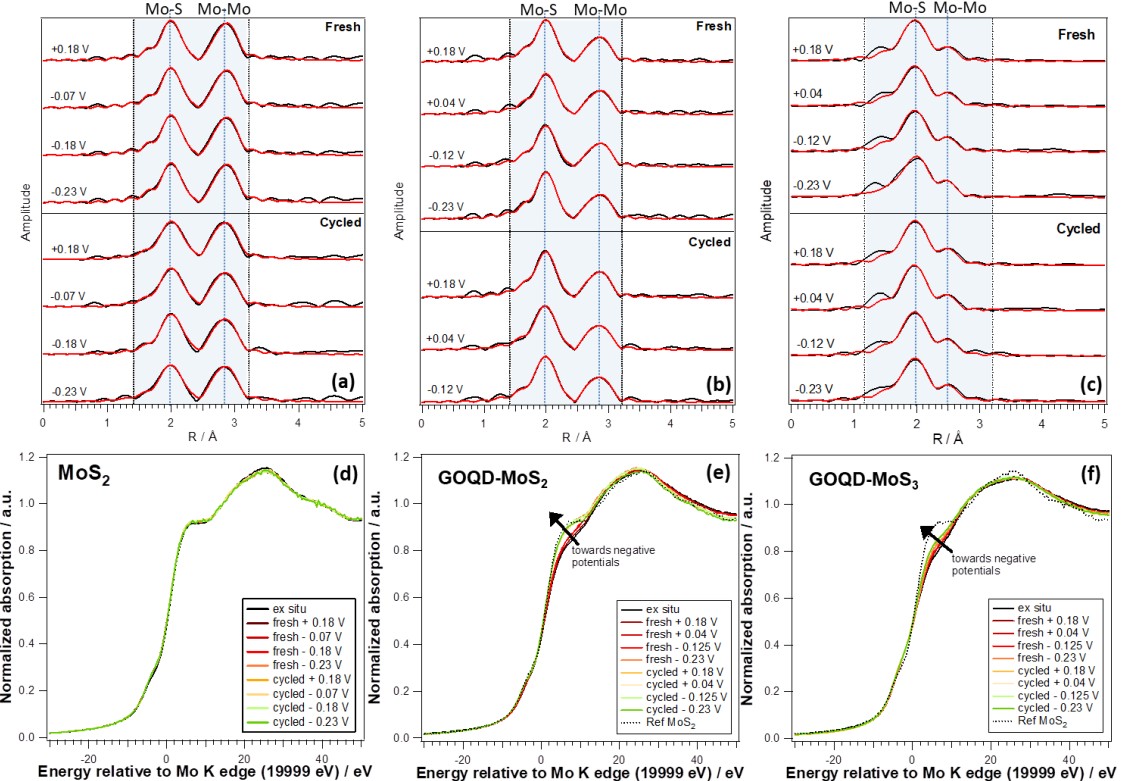

**Figure 3.** Fourier transformed EXAFS (**a,b,c**) and XANES spectra (**d,e,f**) at Mo K edge for the fresh and aged $MoS_2$ (Aldrich) (**a,d**), GOQD-$MoS_2$ (**b,e**) and GOQD-$MoS_3$ (**c,f**) nanohybrids recorded at different applied potentials. The black and red curves represent the experimental data and the corresponding fit, respectively.

The EXAFS data were fitted by using the corresponding $MoS_2$ and $MoS_3$ crystal structures. The $MoS_2$ model consists of a first Mo-S shell with six sulfur atoms at 2.41 Å and a second Mo-Mo shell with six molybdenum atoms at 3.17 Å. For the $MoS_3$ fitting, the Hibble structure [18] was used, although the Weber structure [33] cannot be excluded. In the Hibble model, Mo atoms interact with only one neighboring Mo atom at 2.77 Å and six S atoms at 2.42 Å [8,34]. The fitting results are summarized in Tables S1–S3. The commercial $MoS_2$ was fitted with a Mo-S shell with 5.8 sulfur atoms at 2.41 Å and a Mo-Mo shell with 5.7 atoms at 3.17 Å. These values are very close to those of the bulk (6.0 for Mo-S and 6.0 for Mo-Mo), confirming the large size of the $MoS_2$ particles. Due to the large size, the Mo-O interactions would be negligible. For GOQD-$MoS_2$, $MoS_2$ was fitted with a Mo-S shell with 4.1 sulfur atoms at 2.41 Å and a Mo-Mo shell with 2.5 atoms at 3.17 Å. The small coordination number for the Mo-Mo interaction in this sample is due to the smaller NP size compared with that of the commercial $MoS_2$.

Under electrochemical conditions, $MoS_2$ at +0.18 V (before HER) was fitted with a Mo-S shell with 5.3 sulfur atoms at 2.41 Å and a Mo-Mo shell with 5.1 atoms at 3.17 Å. At more negative potentials (from −0.07 V to −0.23 V vs. RHE), no significant changes were observed in the coordination numbers (*N*) and bond distances (*R*), and either in the disorder parameter ($\sigma^2$), indicating that this material is very stable in this potential range. The fact that no significant changes were observed for this material is also attributed to the large dimension of the NPs and a resultant small fraction of atoms on the surface in contact with the electrolyte.

The best fit for GOQD-MoS$_2$ at +0.18 V was obtained with a Mo-S shell with 4.3 sulfur atoms at 2.41 Å and a Mo-Mo shell with 2.7 atoms at 3.17 Å. In this case, an increase of the coordination numbers ($N_S$ = 5.0 and $N_{Mo}$ = 3.4) was observed when the HER conditions were reached (E = −0.23 V vs. RHE), as well as of the disorder for the Mo-Mo interaction. Taking also into account the X-ray absorption near edge structure (XANES) data reported in Figure 3e, a significant change of the white line at −0.23 V is noted, which shows that the spectral fingerprint matches exactly with that of the MoS$_2$ reference. This effect could be associated with the reduction under HER conditions of the small fraction of MoS$_3$ (seen by XPS in the as-prepared sample) to MoS$_2$ and/or the small amount of oxides present at the edges. The aging treatment did not cause significant changes in *N*, *R* or $\sigma^2$ for the MoS$_2$-based materials, indicating that no major restructuring occurred and, therefore, MoS$_2$ was found to be very stable under HER conditions. This stability is attributed to the high crystallinity of the MoS$_2$ NPs obtained.

For GOQD-MoS$_3$, the best fit at +0.18 V was obtained by a Mo-S shell with 3.8 sulfur atoms at 2.44 Å and a Mo-Mo shell with 1.3 atoms at 2.75 Å, which are in good agreement with those found in the literature for MoS$_3$ [8]. As the potential became more negative, $N_S$ increased while $N_{Mo}$ decreased slightly. In addition, a slight increase in the Mo-Mo bond distance was observed, which could suggest the reduction of the MoS$_3$ NP surface to MoS$_2$. However, the fraction of Mo atoms that change the coordination environment under the HER conditions is too small to observe significant changes in the EXAFS analysis. After the aging treatment, the same trend in the $N_S$, $N_{Mo}$ and $R_{Mo}$ was observed. These results suggest that the stoichiometry of MoS$_3$ at the edges could change gradually under HER conditions, whereas the bulk material remains stable, for this reason, the changes observed are not very significant. This fact is also supported by the variation of the XANES spectra with the applied potential and the aging treatment (Figure 3f). Although in the literature it has been stated that MoS$_3$ is not stable under catalytic conditions, being reduced to MoS$_2$ [7,8], our results suggest that only the edges evolve to a structure similar to that of MoS$_2$ while the bulk remains stable as MoS$_3$.

In order to further investigate the chemical changes experienced by the MoS$_x$ materials during HER, in line XPS/ECl measurements were also performed at two different potentials: (i) before HER (+0.18 V); under HER conditions (−0.4 V). XPS allows us to track the changes produced only on the surface of the MoS$_x$ NPs, where the electrocatalytic reaction takes place, and to correlate them with the results obtained by XAS.

The S 2*p* and Mo 3*d* PE spectra for the GOQD-MoS$_x$ samples before and after the electrochemical measurements are displayed in Figure 4, whereas those for the commercial MoS$_2$ can be found in Figure S4. The analysis of the PE lines is summarized in Table S4. As already described above, the components associated with Mo(IV) and Mo(V) were included in the fit of the Mo 3*d* PE lines, whereas the components related to S$^{2-}$ and bridging S$_2^{2-}$/apical S$^{2-}$ were included in the fit of the S 2*p* PE lines. After the electrochemical treatments, a new component related to the formation of hydroxides on the surface due to the contact with the electrolyte was included in the fit. It should be noticed that the component associated with the formation of hydroxides was not included in the EXAFS fits since its contribution to the overall signal is negligible.

For GOQD-MoS$_2$, no significant changes are observed at +0.18 V, as already seen by XAS. At −0.40 V, the sample is very stable under the catalytic conditions without noticeable changes at this potential. The small discrepancy between the XAS data, which showed a slight reduction of the sample under HER conditions, and the XPS data can be attributed to the fact that in the XPS/EC experiment the sample was prepared in situ and not exposed to air before the electrochemical measurements, therefore, no oxidation of the edges occurred. In the case of GOQDs-MoS$_3$, no significant changes occur at +0.18 V, which can also be attributed to the absence of oxidized edges, as explained for the GOQDs-MoS$_2$ sample. However, at HER conditions (E = −0.4 V), an increase of the component associated with S$^{2-}$ in the S 2*p* line, i.e., the component associated with edges terminated with S-monomers, is observed to the detriment of the component related to the edges terminating with the S-dimer (S$_2^{2-}$). The corresponding increase of Mo(IV) and a decrease of Mo(VI) is observed in the Mo 3d line. The atomic Mo:S ratio calculated from XPS at this potential is 1:2.2. It has to be highlighted that if the edges terminate with

S-monomers, the local stoichiometry is close to $MoS_2$. Therefore, this result is interpreted as a change of the sulfur terminal sites, from S-dimer ($S_2^{2-}$) to S-monomer ($S^{2-}$). This change is only produced at the edges of the nanoparticles, since XAS results indicated that the bulk $MoS_3$ NPs are stable under operating conditions, even after the aging treatment.

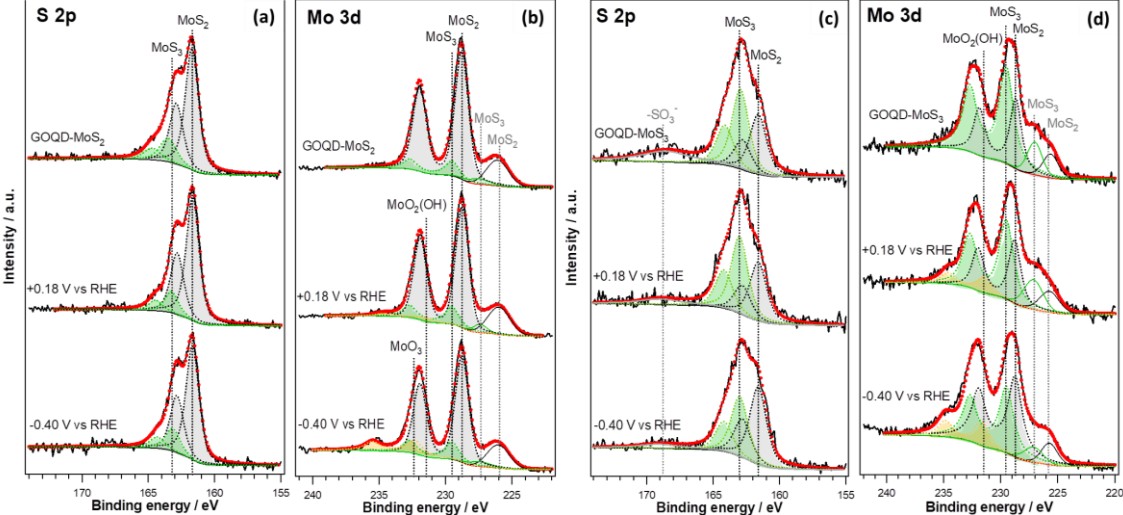

**Figure 4.** S $2p$ (**a**,**c**) and Mo $3d$ (**b**,**d**) photoemission lines and their separation into chemically shifted components for GOQD-$MoS_2$ (**a**,**b**) and GOQD-$MoS_3$ (**c**,**d**) before and after the electrochemical measurements in 0.1 M $HClO_4$ at different potentials. The black and red curves represent the experimental data and the corresponding fit, respectively.

In the literature, different active sites have been proposed; however, our data support the DFT calculations evidencing that the active sites of $MoS_x$ under HER conditions are terminated by S-monomers, confirming the crucial role of S coordination in determining the catalytic activity [7]. Theoretical calculations have also demonstrated that the most favorable mechanism for the hydrogen evolution on $MoS_2$ involves the first proton adsorption at the S-edge, suggesting that this edge is chemically more active than the Mo-edge [10,35,36]. Therefore, this study confirms that the stable phase of $MoS_x$ in HER conditions entails edges terminated with S-monomers, whose stoichiometry is close to $MoS_2$. This could explain why, even though both GOQD-$MoS_x$ nanohybrids are active toward the HER, GOQD-$MoS_2$ exhibit slightly higher activity in terms of overpotential than GOQD-$MoS_3$, due to the S-monomer surface termination.

## 4. Conclusions

We have investigated the chemical stability of GOQD-$MoS_{x, x = 2,3}$ nanohybrids by using *operando* X-ray absorption spectroscopy and combined photoemission/electrochemical in line measurements. Our study shows that the GOQD-$MoS_2$ and GOQD-$MoS_3$ nanohybrids are both active towards the HER and exhibit remarkable chemical stability under working conditions, most likely due to the stabilization provided by the GOQD. In addition, our findings show that the edges in the GOQD-$MoS_3$ nanohybrids (differently than the GOQD-$MoS_2$) undergo a sulfur coordination change under HER conditions, from S-dimer ($S_2^{2-}$) to S-monomer ($S^{2-}$). This transition has a significant role in the activity of these materials and is probably the reason for the similar activity of both $MoS_x$ phases.

In conclusion, our work provides general information about the experimental tools that allow for optimal characterization of electrocatalytic materials, showing that the knowledge gained from the synergistic application of different investigation techniques allows us to address fundamental questions in energy materials and conversion research.

**Supplementary Materials:** The following are available online at http://www.mdpi.com/2571-9637/3/2/17/s1, Figure S1: In situ electrochemical cell, Figure S2: Raman spectroscopy and XRD characterization, Figure S3: $k^3$ weighted experimental data and fit at Mo K edge at different potentials, Figure S4: S 2p and Mo 3d photoemission lines for the commercial $MoS_2$ at different potentials, Tables S1–S3: Structural parameters obtained from fitting the Mo K edge, Table S4: Analysis of the single chemical components of the S 2p and Mo 3d regions at different potentials.

**Author Contributions:** Investigation, L.C., M.F. and M.C.; Data Analysis, L.C. and S.W.T.P.; Conceptualization, L.C., A.E.R. and S.A.; Draft writing, L.C., S.A., G.G.; Supervision, G.G. All authors have read and agreed to the published version of the manuscript.

**Funding:** This work was partially supported by the following projects: Italian MIUR (PRIN, SMARTNESS, 2015K7FZLH, PRIN, MULTI-E, 20179337R7), and MAECI Italy-China bilateral project (GINSENG, PGR00953).

**Acknowledgments:** The authors wish to acknowledge the Diamond Light Source for provision of beamtime (SP9550), and Diego Gianolio is acknowledged for the excellent beamline support.

**Conflicts of Interest:** The authors declare no conflict of interest.

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
