# Peer review of "In Situ Study of Graphene Oxide Quantum Dot-MoSx Nanohybrids as Hydrogen Evolution Catalysts"

_surfaces, doi:10.3390/surfaces3020017_

Round 1

Reviewer 1 Report

The manuscript presents combined XPS/EC and operando XAS characterization to illustrate the role of sulfur terminal sites in MoSx for HER. The research topic is interesting and the experimental results are well interpreted. Hence, I recommend its publication in Surfaces after minor revision.

  1. Figure 1 (b) shows the S2p spectra of GQD-MoS3, however, only one component is fitted for the broad peak around 168 eV. Can the authors explain why? It seems that this peak is related to sulfate species.
  2. The Tafel slops are not compared among the materials investigated. This parameter should be presented and discussed in the manuscript.
  3. I have a strong concern with the authors’ claim that S-monomers are more active than Mo edges based on the presented results. Firstly, as aforementioned, the Tafel slops are not compared, therefore, it’s too ambitious to claim that GQD-MoS2 is more active than GQD-MoS3. Secondly, even though GQD-MoS2 is more active than GQD-MoS3, a direct comparison with Mo-terminated (sulfur-poor) MoSx should be done to draw such conclusion.

Small issues:

  1. In Part 2.1, the authors prepared GDQ-MoSx nanohybrids by dissolving (NH4)2SO4 in GQDs solution. But where is the Mo source?
  2. A zoom-in Raman spectra in Figure 1 (a) should be presented to better interpret the peak assignments.
  3. The assignment of species in XPS can be better presented, for instance, ‘S 2s’ is shown in Figure 1b, but ‘MoS2’ and ‘MoS3’ is used in the S 2s region of Figure 3.

Reviewer 2 Report

Report on the submitted research article Surfaces - 814708         

            This manuscript describes the synthesis and the characterization of graphene quantum dots (GQDs)-MoSx nanohybrids with different MoSx stoichiometries. In particular, the chemical stability and the catalytic activity of the samples were tested in electrochemical cells under hydrogen evolution reaction (HER) by using photoemission/electrochemical (XPS/EC) measurements and operando X-ray absorption spectroscopy (XAS). In addition a series of complementary characterisation techniques such as X-ray photoemission spectroscopy (XPS), X-ray diffraction (XRD) and Raman spectroscopy were used.  The results show that the investigated (GQDs)-MoSx nanohybrids exhibit a remarkable chemical stability under working conditions, most likely due to the stabilization provided by the GQDs. Their higher catalytic activity compared to that of commercial MoS2 is attributed to the increase of the exposed edge sites due to the coupling with GQDs that limit the growth of the MoSx NPs. The stable activity of the nanohybrids is mainly explained by an undergoing sulfur coordination change under HER conditions.

            The manuscript is well written, the references are adequate, and the results well and analytically presented. The conclusions are interesting and useful in the field of the nano-catalysts. Therefore, in my opinion, this work is appropriate for publication in Surfaces. Some following comments and queries may further improve the manuscript.

Majors

  • Title and throughout the paper: Does GQDs mean graphene quantum dots or graphene oxide quantum dots? If the latter, then GOQDs is more suitable.
  • Line 55: What do the authors mean with the ratio Mo(IV)/Mo(V)? Are these different states of Mo oxidation (different valence)?
  • Line 73: When we are talking about sheets and their dimensions, we should give area instead of length. What does 5 nm represent?
  • Line 88: It is not clear how the authors introduced Mo in the electrodes. What substance was used? More information should be provided.
  • Section 2.3: A picture of the electrochemical cell within the manuscript would be helpful. I looked at the ref.15 but the picture of the cell is mentioned as a supplementary file which I never managed to retrieve.
  • Line 113: Please define LSV.
  • Figure 1: I think fig. 1 contains too much information. I suggest to split up the figure at least into two figures. Fig. 1b contains 6 subfigures making difficult any detailed observation. Fig. 1b could stand on its own.
  • Figure 2: Looking carefully at the 2b subfigure, the Mo-Mo distance for the GQD-MoS2 sample is closer to 2.9 than to 2.8 Angtrom. The vertical dotted line is not well aligned to the EXAFS peak.
  • Figure 2: Looking at the EXAFS and XAS measurements for the fresh and aged MoS2 (Aldrich) and GQD-MoS2 respectively, the applied voltages in EXAFS measurements are different than those in XAS. What for this happen?
  • Figure 3: Based on the XPS analysis, in the case of the GQD-MoS3 sample, the component due to the MoS2 is comparable to that of MoS3 (see also Table S4). This shows that the homogeneity as far as it concerns the chemical state of Mo in the GQD-MoS3 sample is not good. Consequently, how fair it is to name the sample as GQD-MoS3?

Minors

  • 8: The correct title of the journal is J. Am. Chem. Soc.
  • Line 39: Please write (PGMs).
  • Line 114: Please don’t define HER again (see introduction).
  • Line 100: Please write MoS2(100).
  • Line 208: Bigger instead of higher.
  • Ref. 33: Please write C. Muijsers.
